# Paternity success for resident and non-resident males and their influences on paternal sibling cohorts in Japanese macaques (*Macaca fuscata*) on Shodoshima Island

**Shintaro Ishizuka**[1,2,3,4] *, **Eiji Inoue**[2], **Yuki Kaji**[5]

1 Faculty of Life Science and Technology, Department of Biological Science, Fukuyama University, Fukuyama, Hiroshima, Japan, 2 Faculty of Science, Department of Biology, Toho University, Funabashi, Chiba, Japan, 3 Japan Society for Promotion of Science, Kojimachi Business Center Building, Chiyoda-ku, Tokyo, Japan, 4 Primate Research Institute, Kyoto University, Inuyama, Aichi, Japan, 5 Choshikei Monkey Park, Tonosho, Kagawa, Japan

* ishizuka.shintaro@fukuyama-u.ac.jp

**Data Availability Statement:** All relevant data are within the paper and its Supporting Information files.

## Abstract

Reproductive success can be attributed to both resident and non-resident males in non-human primates. However, reproductive success of non-resident males has rarely been investigated at an individual level. As resident males achieve different degree of reproductive success with regard to various factors, such as male dominance relationships or female mate choice, the degree of reproductive success for non-resident males may vary between individuals. As male reproductive success is highly skewed towards specific individuals, the percentage of similar-aged paternal siblings within groups is expected to increase. However, the extent to which each male contributes to the production of cohorts of paternal siblings remains unclear. Here we examined the paternity of offspring born over five consecutive years in a free-ranging group of Japanese macaques *Macaca fuscata* on Shodoshima Island, Kagawa Prefecture. Genotypes of 87 individuals at 16 autosomal microsatellite loci were analyzed and paternity of 34 offspring was successfully assigned to a single candidate father. We quantitatively assessed paternity success for resident and a few non-resident males whose genetic samples were successfully collected. We quantitatively assessed the percentages of paternal siblings in the same age cohorts produced by those males. Non-resident males sired similar percentage of offspring compared to resident males. A large prime-aged non-resident male was the most successful sire among males in two of the five years. These results provide new insights that male reproductive success could be highly skewed toward a specific non-resident male. Subadult males had a lower percentage of paternity success, which may be because females may prefer physically mature males. Various males, including non-resident males, contributed to the creation of paternal sibling in the same age cohort. The overall results highlighted that not only resident but also non-resident males play an important role in shaping within-group kin structures.

**Funding:** This study was financially supported by the JSPS KAKENHI (21J00922 and 22K15191 to SI), Cooperative Research Program of the Wildlife Research Center, Kyoto University (2020-B-05 to SI), and Leading Graduate Program in Primatology and Wildlife Science, Kyoto University. The funders had no role in study design, data collection and analysis, decision to publish, or preparation of the manuscript.

**Competing interests:** The authors have declared that no competing interests exist.

## Introduction

Male competition over access to females plays important roles to secure breeding opportunities in animals [1,2]. As a consequence of male–male competition, males often achieve different degrees of reproductive success. In group-living primates, variance in reproductive success among males is sometimes explained by dominance relationships [3–6]. Studies have shown that reproductive success is often skewed toward high-ranking males within multimale groups [7–13]. Beside dominance relationships, female mate choice also can affect the distribution of male reproductive success [14,15]. Females may prefer to mate with particular categories of males, depending on the characteristics of males, such as their maturity, capacity for investment, or genetic background [16,17]. Males that are selected by females can achieve high reproductive success [18–22].

Variance in male reproductive success has been investigated within social groups. Yet, reproductive success for males outside a group has been infrequently examined. In group-living animals, especially primates, both resident and non-resident males have opportunities to breed with females of the group. The paternity of offspring assigned to males outside a group is termed extra-group paternity (EGP). EGP is a phenomenon widely observed in various primate species [23–28] and is essential for the evolution of social systems in animals as it increases genetic diversity within groups and is consequently associated with the increased chance of offspring survivorship [29]. EGP has been usually determined when the paternity of offspring is unsuccessfully assigned to all candidate fathers within a group, probably because individual identification and sample collection for non-resident males are usually difficult in the field. Several studies that have examined the overall percentages of EGP have shown that their percentages would increase when the number of males within groups decreases or when female reproductive synchrony increases [26,30,31]. However, few studies have assigned EGP to a particular non-resident male, and the number of cases in which paternity success for each non-resident male has been assessed at the individual level is scarce [32]. Similar to the pattern in resident males, the degree of reproductive success for each non-resident male may vary depending on various factors, such as their dominance relationships or female choice. To understand mechanisms that influence the variance in the male reproductive success among all types of males, it is necessary to assess the reproductive success for not only the resident males, but also the non-resident males, by individually identifying and sampling both.

Variance in the male reproductive success modulates the abundance of paternal kin living in animal groups. The presence of kin within groups plays important roles for their social lives because kinship is one of the key factors that affects affinity among individuals, and reciprocity and cooperation are attributed to kin selection [33–35]. There is evidence of paternal kin bias behavior in primates [36–39]. The evidence suggests that primate individuals have more social partners as the number of paternal kin increases in their groups. Theoretically, the percentage of paternal sibling dyads in the same age cohort is expected to increase with a higher male reproductive skew [40–42]. Since male reproductive success is often skewed toward the highest-ranking males in primate groups, researchers have assessed the percentage of paternal sibling dyads in the same age cohort according to the percentage of paternity success by the highest-ranking male [38,42,43]. However, the paternity success can be skewed towards other resident males or non-resident males besides the highest-ranking male. The contributions of those non-highest-ranking males to the percentages of paternal sibling dyads in the same age cohorts have not been investigated. For a comprehensive understanding of the mechanisms that produce kin-dyads within animal groups, it is crucial to examine the percentages of paternal sibling dyads produced by non-resident males at an individual level.

The Japanese macaque (*Macaca fuscata*) provides an interesting opportunity for the investigation of paternity success of non-resident males and their contributions to the production of within-group kin-dyads. They form female-philopatric multi-male and multi-female groups characterized by strong matrilineal affiliative relationships and a polygynandrous mating system [44–51]. Males typically emigrate from their natal group when they are approximately four or five years old [52] and continue to transfer between groups throughout life [53]. The tenure of males in a group is approximately three years [54]. Dominance rank is linear and stable in both sexes [54–56], although troop takeovers by non-troop males were reported from the Yakushima population [57]. Non-resident males often breed with females of the group, as the percentages of EGP are relatively high among primates [28,31]. This is partially because they have breeding seasonality [58,59], and novel and unfamiliar males attain high reproductive success because they are strongly selected by females for mating partners when females are likely to conceive [19,21,60,61]. Further, the exact number of siring for each non-resident male in the field can also be assessed. During the mating season, several non-resident males sporadically appear around the group and copulate with the group's females [24,62]. Considering that previous field research succeeded in collecting behavioral data and genetic samples from non-resident males [19,24,63], it is possible to expand on it further to examine the paternity success for non-resident males at the individual level. Moreover, the percentage of paternity by the highest-ranking male is usually low [19,24,64], although one study reported a relatively high percentage [65]. This suggests that the percentages of paternal sibling dyads produced by non-resident males can be assessed quantitatively.

In this study, we examined the paternity of the offspring born in five consecutive years within a group of Japanese macaques, wherein resident and a few of non-resident males were identified and sampled. We assessed the percentages of paternity success for both resident and non-resident males and the percentages of paternal sibling dyads produced by them among the offspring in each year.

## Materials and methods

### Study subjects

The study subjects were free-ranging Japanese macaques of the B-group inhabiting an area near the Choshikei Monkey Park on Shodoshima Island [51]. The macaques are provisioned by park staff at approximately 8:30 h, 14:00 h, and 16:00 h and by tourists visiting the park. We initially conducted field research on the subjects and started individual identification of the group between September and December 2017. In April 2018, we started monitoring the presence of identified individuals in the monkey park on a daily basis (five or six days per week), although their presence has not been systematically recorded. In February 2019, we successfully completed individual identification of group members, and started systematically recording the daily presence of the members of the group. The mean number of observation days per month for 2019–2022 was 25.6. The group size, the number of adult females, and the number of adult and subadult males for 2019–2021 are shown in Table 1.

### Classification of males

This study focused on males estimated to be aged ≧ 4 years and sexually mature [66]. They were classified based on their age class and social status. Following a previous study [19], the age class of males aged 4–6 years and that > 6 years was categorized as "subadult" and "adult," respectively.

For this study, the males were first divided into resident males of the study group or non-resident males according to the daily monitoring. After completing individual identification of

**Table 1. Composition of the B-group at the end of 2019–2021.**

|  | 2019 | 2020 | 2021 |
|---|---|---|---|
| Adult females | 29 | 16 | 16 |
| Adult and subadult males | 12 | 13 | 11 |
| Juveniles and infants | 40 | 35 | 33 |
| Group size | 81 | 64 | 60 |

The number of group members gradually decreased from 2019 to 2020. Although the reason for the decline remains unclear, this might be because several members were captured by the government or local people.

he study group in February 2019, we sometimes encountered unidentified males around the mating season. Those males were seemingly unhabituated to human observers and did not eat provisioned food, implying that they were not members of the group. We regarded those males as "non-resident males". We identified four non-resident males during the study period.

The social status of the resident males was further divided into dominant or subordinate males. Males of provisioned groups constantly compete with one another over food, which emphasizes the dominance-subordinate aspects of their relationships [67]. In our study, some males were allowed to access preferred food at the provisioning area, whereas other males were driven away by both the males and females who stayed at the provisioning area and were thus uncommonly observed to enter into the area. Therefore, we classified those males that were allowed to access the provisioning area as dominant and those that were driven away from the provisioning area as subordinate males. Collectively, the social status of males was classified into three categories: dominant, subordinate, and non-resident males. This is in accordance with previous studies on macaque species [24,68]. Two to three dominant, eight to ten subordinate, and two identified non-resident males were present between 2019 and 2021, respectively. The presence and social/age class of all identified males in each mating/non-mating season is shown in S1 Table. We regarded September–December as the mating season, and January–August as the non-mating season because births occur between March and June [58,59], and the mean gestation length of this species is 176 days [69].

## Genetic analysis

We used genotype data for the study subjects at 16 autosomal microsatellite loci, which were obtained in our previous work [51]. The study estimated dyadic genetic relatedness between females within the study group to assess the strength of kin relationships between them. Since this study newly assessed paternity success for resident and non-resident males, this study does not constitute dual publication. Feces, sperm, and saliva samples were collected using cotton swabs and stored in lysis buffer at ambient temperature immediately after we observed defecation, ejaculation, and materials the animals sucked in the field. DNA was extracted from the samples using a QIAamp Stool Mini Fast Kit (Qiagen, CA, USA). Using DNA extracts, the genotypes at 16 autosomal microsatellite loci were analyzed using the multiplex PCR method [70]. Amplification products were separated by capillary electrophoresis using an ABI 3130xl Genetic Analyzer (Applied Biosystems, CA, USA). Alleles were sized using Peak Scanner (Applied Biosystems). Since DNA extracted from non-invasively collected samples is typically degraded and low in concentration [71,72], we repeated the genotyping to ensure accuracy following the criteria in the previous study [70]. Details for the methods of genetic analysis are described in a previous study [51].

## Paternity analysis

The paternity of 46 offspring (8 of 18 in 2018, 9 of 19 in 2019, 8 of 13 in 2020, 10 of 10 in 2021, and 11 of 11 in 2022, respectively) was investigated. The offspring whose paternity was not investigated were already missing and no genetic samples had been collected from them. Paternity analysis was conducted using the pairwise likelihood approach with CERVUS [73]. The resident candidate fathers were only fully sampled for the offspring born in 2020, 2021, and 2022. Therefore, the percentage of EGP could be calculated for only these years. According to our field observations, there seemed to be approximately 10 unsampled candidate fathers who belonged to the group during the 2017 and/or 2018 mating seasons or were non-resident males. Since we collected genetic samples for 19 candidate fathers, which included three of the four identified non-resident males and accounted for 60–70% of all candidate fathers, the proportion of sampled candidate fathers was eventually assumed to be 0.60. The proportion of loci mistyped and the error rate in the likelihood calculation was both 0.01 because genotype accuracy was confirmed following reasonable criteria [70]. Paternity for 42 of the 46 offspring was analyzed under the condition that the offspring's maternity was known since genotypes for the mothers were determined. However, genetic samples for mothers of four offspring could not be collected. Therefore, the paternity for the four offspring was analyzed under the condition that the maternity of the offspring was unknown. When the most likely father had no mismatched alleles and the confidence level for the assignments was more than 95%, the male was concluded to be the offspring's father.

## Statistical analysis

To analyze the likelihood of siring by each male, we constructed generalized linear mixed models (GLMMs) with binomial distribution and logit link function. We included the percentage of sires for each male in each year as a response variable. Since the total number of sampled males was 10, 10, 12, 12, and 13 in 2017, 2018, 2019, 2020, and 2021, respectively, the total number of data points was 57. The response variable ranged from 0 to 0.40, as the most successful sire's share in a year was 40% (see the Results). The social status (dominant/subordinate/non-resident) and age class (adult/subadult) of each male and the number of newborns, which corresponds to the approximate number of estrus capable females in each year (18 in 2018, 19 in 2019, 13 in 2020, 10 in 2021, and 11 in 2022), were included as predictor variables. Offspring ID and the study year were included as random effects. We calculated the Akaike's information criterion (AIC) for the constructed model and null model. Comparing the AIC values, we assessed the fit of constructed model. The models were fit using the "lmer" function of the R package lme4 [74].

## Ethics statement

This study was permitted by the Choshikei Monkey Park on Shodoshima Island, Japan. All methods were designed to be noninvasive for the subjects. Field research was conducted in accordance with the American Society of Primatologists Code of Best Practices for Field Primatology, and conformed to the Guidelines for Field Research established by the Ethics Committee of the Primate Research Institute of Kyoto University. All aspects of this study adhered to the ASAB/ABS Guidelines for the use of animals in research.

## Results

### Paternity

Genotypes of the 88 individuals at 16 autosomal microsatellite loci were analyzed (S2 Table). The number of alleles, heterozygosity, and allelic dropout rate are investigated based on the

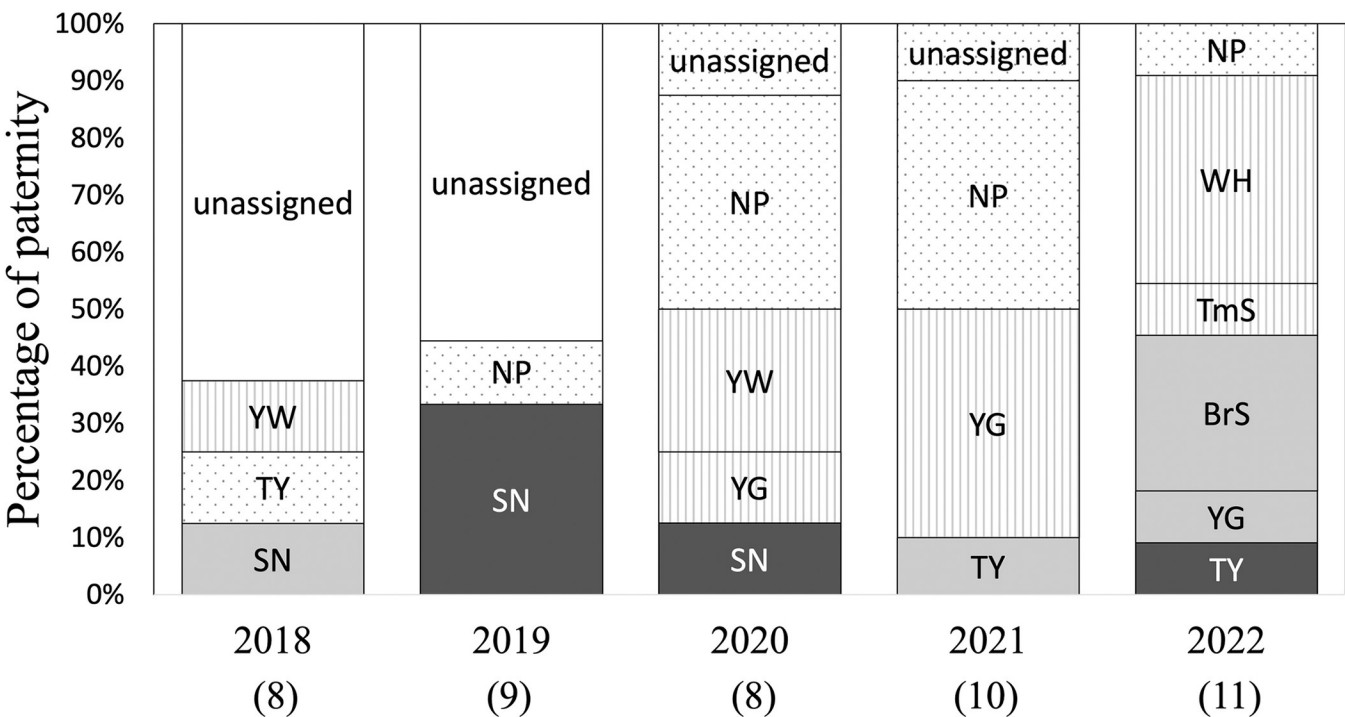

**Fig 1. The number of offspring sired by each male in the five-year study period.** Gray, lined, and dot bars indicate sires by the dominant, subordinate, and non-resident males, respectively. White bars indicate sires whose father's social status remained unclear. The ID of all fathers is shown in each bar. The values within brackets represent the number of paternities analyzed in each year.

genotypes (S3 Table). The paternity of 34 of the 46 offspring was assigned to a single candidate father (Fig 1 and S4 Table). At least one offspring was sired by sampled non-resident males in each year. Since all resident candidate fathers for offspring born in 2018 and 2019 were not sampled, it remains unclear whether or not these offspring were sired by non-resident males, and thus, the percentage of EGP could not be calculated for those years. However, for the years in which all resident candidate fathers were sampled, the results revealed that the percentage of EGP was 50% (4 of 8) in 2020, 50% (5 of 10) in 2021, and 9% (1 of 11) in 2022. The overall percentage of EGP between 2020 and 2022 was 34% (10 of 29). Interestingly, 75% (3 of 4) and 80% (4 of 5) of the EGP cases were assigned to a single non-resident male "NP" in 2020 and 2021, respectively. He was the most successful sire and sired 38% (3 of 8) and 40% (4 of 10) of offspring in 2020 and 2021, respectively. The proportion of paternity success for each male across the study years is shown in S5 Table.

### Effects of male categories on the percentages of paternity success

The constructed model was significantly better fit than the null model ($\Delta$AIC = 10.99, $P < 0.001$). The constructed model showed that the effects of neither male social status nor the number of estrus capable females were significant (Fig 2A and Table 2). The effect of male age class had a negative effect on the percentage of paternity success, indicating that paternity success of subadult males was lower than that of the adult males (Fig 2B and Table 2).

### The proportion of paternal sibling dyads

The percentages of paternal sibling dyads among the offspring born in 2018, 2019, 2020, 2021, and 2022 were 0%, 8%, 14%, 27%, and 16%, respectively (Table 3). The overall percentage of

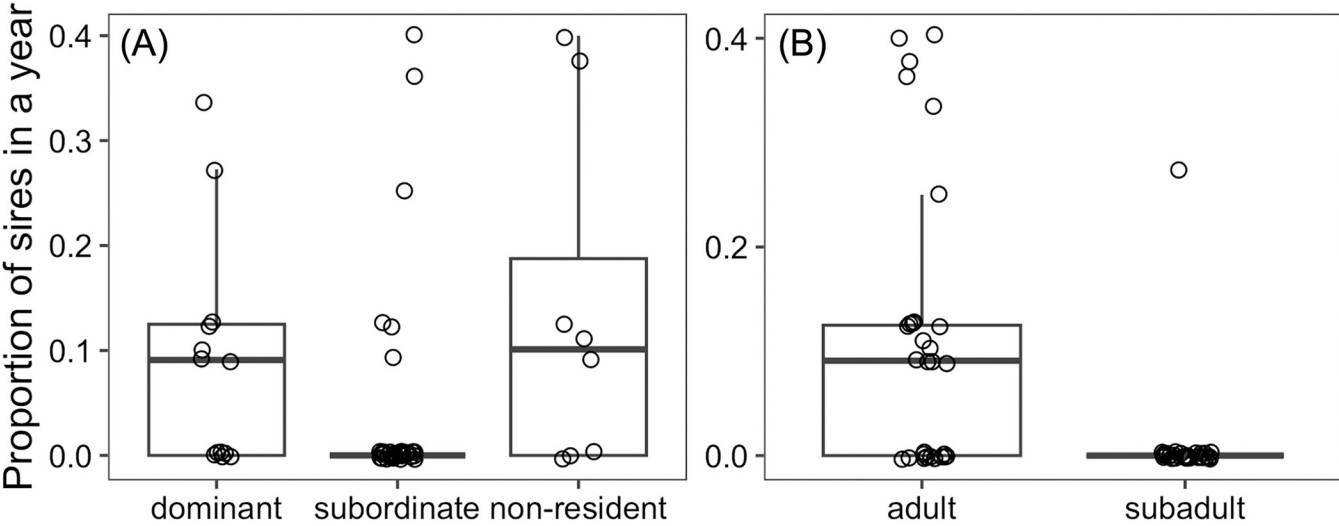

**Fig 2. The effects of males' social status and their age class on their paternity success.** Proportion of paternity success by each male in each year according to the males' social status (A) and their age classes (B) are shown. Boxes indicate the first to third quartile of observed values, solid lines show the median, and each dot represents the proportion of paternity success by a sampled candidate father in each year.

paternal sibling dyads was 15%. Interestingly, subordinate males or non-resident males produced 11–13% of paternal sibling dyads between 2020 and 2022.

## Discussion

The percentage of EGP between 2020 and 2022 was 34% (9–50%) in the Shodoshima B-group. These results are consistent with previous results showing that 23–80% of offspring were sired by non-resident males (Table 4). The percentage of EGP in this species is relatively high among primate species [28,31]. One reason for the relatively high percentage of EGP may be the high extent of female receptive synchrony in this species [65]. EGP is more likely when the number of males within groups decreases and when female reproductive synchrony increases [26,30,31]. Similar to other species that exhibit both high female receptive synchrony and EGP, resident males may not guard receptive females against non-resident males.

A non-resident male, "NP," was the most successful sire in both 2020 and 2021. Our results highlighted that at least some proportion of non-resident males could have chances to gain a high percentage of paternity success, although the reproductive success of non-resident males as a whole cannot be assessed because of the incomplete sampling of non-resident males. Given that the non-resident males sampled in this study might have chances to breed with females of other groups beside our study group, the percentage of paternity success for these non-resident males might have been underestimated. Since female Japanese macaques prefer

**Table 2. Results of the generalized linear mixed model (GLMM) performed to test the effects of males' social status and their age class.**

| Parameter | Log-Odds | SE | 95% CI | Z | P |
|---|---|---|---|---|---|
| (Intercept) | −1.60 | 1.10 | −3.76 to 0.55 | −1.46 | 0.145 |
| Male category (non-resident vs. dominant) | 0.47 | 1.09 | −1.67 to 2.60 | 0.43 | 0.669 |
| Male category (non-resident vs. subordinate) | 0.11 | 0.75 | −1.37 to 1.58 | 0.14 | 0.888 |
| Age (adult vs. subadult) | −3.00 | 1.05 | −5.06 to −0.95 | −2.86 | < 0.01 |
| No. of capable females | −0.08 | 0.06 | −0.21 to 0.04 | −1.26 | 0.208 |

**Table 3. The percentage of paternal sibling dyads among offspring born in each year. The values within the brackets represent the number of paternal sibling dyads in each category.**

| Year | No. of dyads analyzed | Percentage of paternal sibling dyads | Maximum percentage of paternal sibling dyads produced by one male | Male category of the most successful sire |
|------|------|------|------|------|
| 2018 | 28 | 0(0)* | N/A | N/A |
| 2019 | 36 | 8(3)* | N/A | N/A |
| 2020 | 28 | 14(4) | 11(3) | NR |
| 2021 | 45 | 27(12) | 13(6) | S and NR |
| 2022 | 55 | 16(9) | 11(6) | S |
| Total | 192 | 15(28) | 12(15) | - |

S and NR represent subordinate and non-resident males, respectively. N/A represents data that was not available.

* The percentages may have been underestimated because offspring whose fathers remain unidentified may have been paternal siblings.

to mate with unfamiliar males [75,76], non-resident males may maintain unfamiliarity with females for at least several years, and consequently have several chances to breed with females. Furthermore, our results provide new insights that there is a reproductive skew among non-resident males; 75–80% of EGP were attained by "NP" in 2020 and 2021, while "TN" was unsuccessful in siring offspring in those two years. According to our field observations, "NP" was a large prime-aged male and displayed frequently (Fig 3). A previous study in captivity showed that female Japanese macaques expressed mate choice behavior toward males who displayed most frequently when they were likely to conceive [61]. Since non-resident males are typically unfamiliar to females and the extent of unfamiliarity to females might not differ between them, a non-resident male who displayed frequently might be favored as females' mating partner and gain higher paternity success.

Notably, subadult males sired a lower percentage of offspring than the adult males. These results were consistent with previous studies showing that males reach high reproductive

**Table 4. Overview paternity results in Japanese macaques.**

| Site | Number of adult males | Number of adult females | Condition of study site | Period | Number of paternity | Maximum percentage of paternity for one male (%) | Note for the status of the most successful sire | Overall EGP (%) | Reference |
|------|------|------|------|------|------|------|------|------|------|
| Kyoto University | 19 | 30 | Captive | 1988 | 48 | 22.9 | Highest-ranking | - | [64] |
| Yakushima, Nina-A | 15 | 15 | Wild | 1998 | 9 | 22.2[b] | Highest-ranking | 33 | [65] |
| Yakushima, Nina-A | 7–10 | 2–7 | Wild | 1999–2000 | 4 | 25 | Low-ranking, and non-resident | 25 | [24] |
| Yakushim, B | 1–7 | 6–7 | Wild | 1996–2000 | 5 | 20[b] | High-ranking, low-ranking, and non-resident | 80 | |
| Arashiyama | 24–27 | 93–100 | Free-ranging | 2002–2003 | 23 | 15.3 | Low-ranking | 23 | [19] |
| Oregon | 17 | N/A | Semi free-ranging | 2018–2019 | 34 | 20.5 | High-ranking[c] | - | [21] |
| Shodoshima[a] | 11–13 | 16–29 | Free-ranging | 2019–2021 | 46 | 27.5 | Non-resident | 34 | Present study |

a The data in 2018 and 2019 were excluded because of incomplete individual identification.

b The percentage may become higher if EGP is attributed to the single male.

c It remains unclear whether the male was the highest-ranking male.

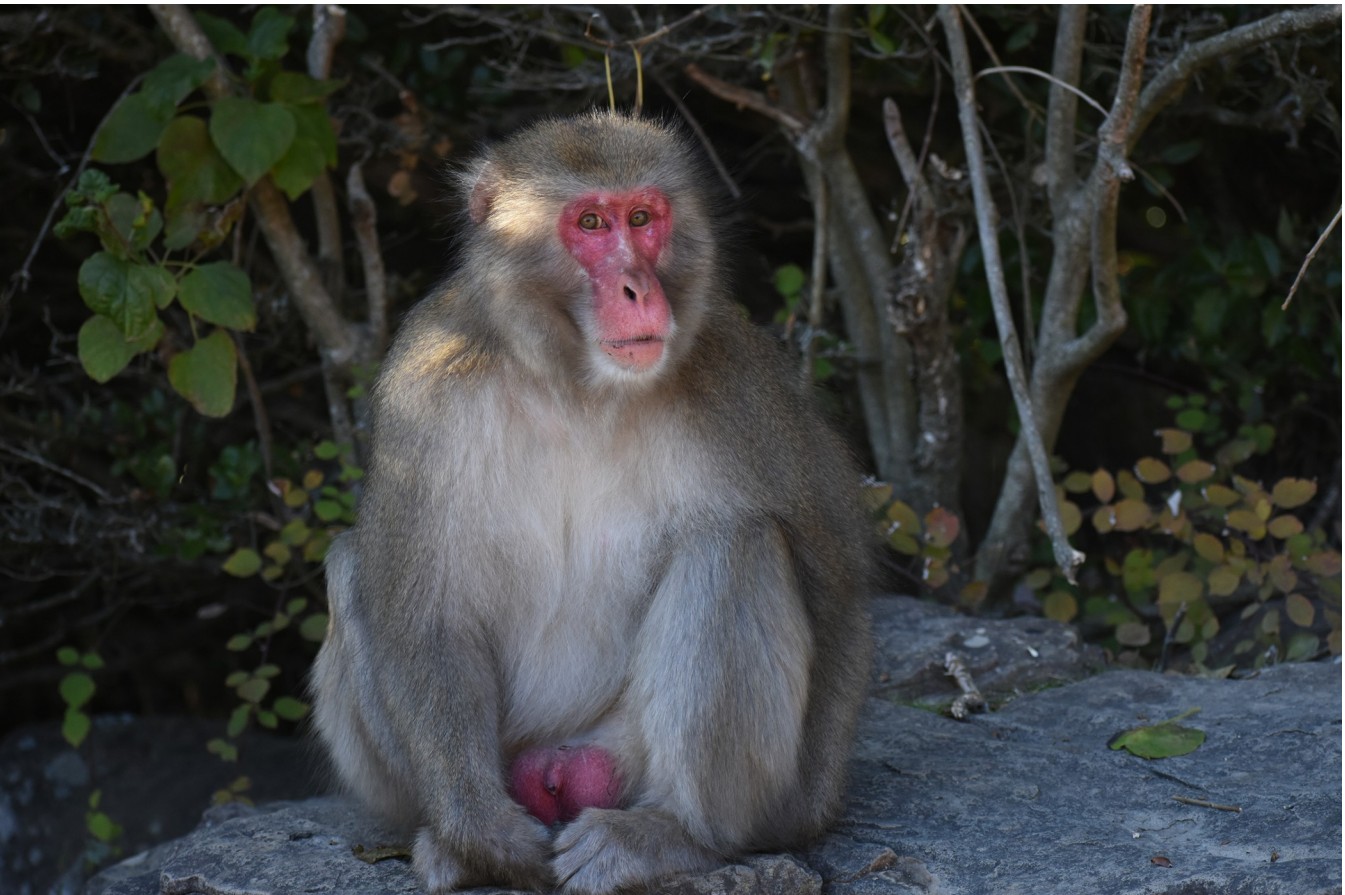

**Fig 3. A photograph of the non-resident male who attained a high percentage of paternity success.** His individual ID was "NP".

success when they are approximately six years old in rhesus macaques *Macaca mulatta* [77,78], and subadult males (aged 4.5–6.5 years old) had a much lower reproductive success than adult males (7.5–25 years old) in Barbary macaques *Macaca sylvanus* [79]. The tendency in which males who reach an age of approximately six years old can gain high amounts of reproductive success may be common among macaque species, although all of these results were observed in provisioned or semi-closed environments. Age-related variance in male reproductive success may be influenced by a function of the interaction between social skills and morphological traits [80,81], or facial coloration of males [82–84].

The percentage of paternal sibling dyads among offspring in the same age cohort was 15% (8–27%) in the Shodoshima-B group. Although available data for comparisons with our results is scarce, the percentage is relatively high compared to 12% in rhesus macaques [42] and 5% in Assamese macaques *Macaca assamensis* [38]. As shown in Table 4, the most successful sires were the highest-ranking males in only two or three of the seven cases. These results suggest that various males, including non-resident males, contribute significantly to the production of paternal sibling dyads in Japanese macaques. The presence of paternal sibling dyads in the same age cohort may trigger kin selection and enhance affinity among individuals of same age. In female-philopatric primates, females of same or similar age often form strong affiliative relationships (e.g. brown capuchin monkeys *Cebus apella nigritus*: [85]; chacma baboons *Papio hamadryas ursinus*: [86]; rhesus macaques: [87]). In Japanese macaques on the Katsuyama

population, the two subject females of the same age formed long-term grooming partnerships [49]. Such an affinity among same or similar-aged individuals in Japanese macaques may be favored by paternal sibling relationships, although it is unclear whether individuals can discriminate paternal kin. Future behavioral and genetic studies are required to clarify the presence or absence of paternal kin bias in this species.

Our findings contribute to a better understanding of how male social status influences diverse reproductive strategies. Notably, this study clearly showed that at least some non-resident males can attain high reproductive success. Furthermore, various males, including non-resident males, significantly contribute to the production of paternal sibling dyads in the same age cohorts, suggesting that not only the resident males, but also non-resident males play an important role in shaping within-group kin structures. However, future studies are required to examine how paternal siblings interact with each other.

## Supporting information

**S1 Table. Information for candidate fathers.**
(DOCX)

**S2 Table. Genotype data for 88 individuals of the B group.**
(XLSX)

**S3 Table. Summary for population genetic parameters.**
(DOCX)

**S4 Table. Results for paternity of offspring in the B group.**
(DOCX)

**S5 Table. Proportion of paternities for each male across the study years.**
(XLSX)

## Acknowledgments

We thank Dr. K. Watanabe, Mr. A. Nishio, Ms. C. Saeki, Mr. M. Ishii, Ms. K. Miyashita, and Mr. K. Hida for their help with the fieldwork. English language editing was provided by Editage (https://www.editage.jp).

## Author Contributions

**Conceptualization:** Shintaro Ishizuka.

**Data curation:** Shintaro Ishizuka, Yuki Kaji.

**Formal analysis:** Shintaro Ishizuka.

**Funding acquisition:** Shintaro Ishizuka.

**Investigation:** Shintaro Ishizuka, Yuki Kaji.

**Methodology:** Shintaro Ishizuka.

**Project administration:** Shintaro Ishizuka.

**Resources:** Eiji Inoue.

**Supervision:** Eiji Inoue.

**Writing – original draft:** Shintaro Ishizuka.

**Writing – review & editing:** Shintaro Ishizuka.

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
