## [Decision Letter · Decision Letter 0]

30 Jul 2024

PONE-D-24-26360Paternity success for resident and non-resident males and their influences on paternal sibling cohorts in Japanese macaques on Shodoshima IslandPLOS ONE

Dear Dr. ISHIZUKA,

Thank you for submitting your manuscript to PLOS ONE. After careful consideration, we feel that it has merit but does not fully meet PLOS ONE’s publication criteria as it currently stands. Therefore, we invite you to submit a revised version of the manuscript that addresses the points raised during the review process.

**I have carefully read the revised manuscript which I find very interesting and have no further comments. If you can attend to the reviewers' points I am confident I will be able to accept the manuscript.**

We look forward to receiving your revised manuscript.

Kind regards,

Maria Santacà

Academic Editor

PLOS ONE

Journal Requirements:

JSPS KAKENHI (grant numbers 22K15191 to SI), the Cooperative Research Program of the Wildlife Research Center, Kyoto University (2020-B-05 to SI)

We thank Dr. K. Watanabe, Mr. A. Nishio, Ms. C. Saeki, Mr. M. Ishii, Ms. K. Miyashita, and Mr. K. Hida for their help with the fieldwork. This study was financially supported by the Japan Society for the Promotion of Science Grant-in-Aid for JSPS fellows (21J00922 to SI), Cooperative Research Program of the Wildlife Research Center, Kyoto University (2020-B-05 to SI), and Leading Graduate Program in Primatology and Wildlife Science of Kyoto University. English language editing was provided by Editage (https://www.editage.jp).

JSPS KAKENHI (grant numbers 22K15191 to SI), the Cooperative Research Program of the Wildlife Research Center, Kyoto University (2020-B-05 to SI)

4. We noted in your submission details that a portion of your manuscript may have been presented or published elsewhere. Some of genetic data in this study were used in our previous work (Ishizuka & Inoue 2023) for different purposes. Please clarify whether this [conference proceeding or publication] was peer-reviewed and formally published. If this work was previously peer-reviewed and published, in the cover letter please provide the reason that this work does not constitute dual publication and should be included in the current manuscript.

Reviewers' comments:

Reviewer's Responses to Questions

**Comments to the Author**

1. Is the manuscript technically sound, and do the data support the conclusions?

Reviewer #1: Yes

Reviewer #2: Yes

2. Has the statistical analysis been performed appropriately and rigorously? 

Reviewer #1: Yes

Reviewer #2: Yes

3. Have the authors made all data underlying the findings in their manuscript fully available?

Reviewer #1: Yes

Reviewer #2: Yes

4. Is the manuscript presented in an intelligible fashion and written in standard English?

Reviewer #1: Yes

Reviewer #2: No

5. Review Comments to the Author

**Reviewer #1:** I was enthusiastic in my reading of the original manuscript, and my enthusiasm has only grown with these careful and substantial revisions. I would like to sincerely thank the authors. There is nothing more frustrating than peer-reviewing a manuscript only to have your comments ignored. I appreciate the respect and thoroughness with which the authors addressed my comments and suggestions. I look forward to seeing this paper published with the full fanfare and acknowledgement it deserves. My remaining (minor) comments are below:

1. I would include the scientific name in the title of your article

2. I would combine the first two sentences of your introduction to read something like "Male competition over access to females is central to securing breeding opportunities in many species".

3. I would replace the word "animal" with "social" at L65

4. The paragraph from L65-L85 is great!! Really concise but clear explanation of the knowledge gap your study is filling.

5. I would revise the sentence at L89-90 to read "There is evidence of paternal kin-biased behavior in primates" for clarity.

6. I would revise the sentence starting at L102 to read something like "Japanese macaques (Macaca fuscata) provide an interesting opportunity for the investigation of paternity success of non-resident males and their contributions to the production of within-group kin-dyads"

7. L104 Is "bisexual" really the word you're looking for? If you're referring to multiple males and females mating with each other, you might be looking for the word "polygynandrous".

8. I would generally rephrase L104 to read "They form female-philopatric multi-male and multi-female groups characterized by strong matrilineal affiliative relationships and a polygynandrous mating system"

9. L110 "Noteworthily" reads a bit odd to me. I would remove.

10. Has a high percentage of EGPs been previously recorded for Japanese macaques specifically? Or is L110-111 referring to other species?

11. Thank you for expanding table 1 :)

12. Methods overall are greatly improved and offer better detail and clarity!

13. I have no comment on genetic methods as this is not my area of expertise.

14. Very interesting results. The discussion appropriately contextualizes the results within the limitations of the study. I appreciate the comparisons to other specific studies and species.

Great work, authors!

**Reviewer #2: **Thank you for the opportunity to review this interesting manuscript. The authors set out to examine the reproductive success of non-resident:resident male Japanese macaques in a provisioned, habituated population. There is little information on non-residents in the published literature, so this represents a unique and important contribution to the literature. Additionally, the authors calculated the number of paternal siblings in the study group, which will aid in our understanding of how kinship influences social behavior. The authors have revised the manuscript and addressed the previous two reviewers' concerns in my judgement. I recommend publication following minor revision. I used track changes in the attached document to help the authors find places where I changed sentences or had questions regarding their meaning. I wish them the best of luck with future work at this site.

6. PLOS authors have the option to publish the peer review history of their article (what does this mean?). If published, this will include your full peer review and any attached files.

Reviewer #1: No

Reviewer #2: No

---

## [Author Response · Author response to Decision Letter 0]

1 Aug 2024

Academic Editor: I have carefully read the revised manuscript which I find very interesting and have no further comments. If you can attend to the reviewers' points I am confident I will be able to accept the manuscript.

> We sincerely thank your review and handling of our manuscript. We have revised our manuscript based on comments from the reviewers. We have also checked and modified the reference list to fit the style of this journal.

Reviewer #1: I was enthusiastic in my reading of the original manuscript, and my enthusiasm has only grown with these careful and substantial revisions. I would like to sincerely thank the authors. There is nothing more frustrating than peer-reviewing a manuscript only to have your comments ignored. I appreciate the respect and thoroughness with which the authors addressed my comments and suggestions. I look forward to seeing this paper published with the full fanfare and acknowledgement it deserves. My remaining (minor) comments are below:

> We sincerely thank your review and helpful comments on our manuscript. We feel that our manuscript has been improved thanks to your comments. Based on your comments, we have revised our manuscript. We have responded to each of your comments in detail as follows.

1. I would include the scientific name in the title of your article

> As you suggested, we have added the scientific name in the title.

2. I would combine the first two sentences of your introduction to read something like "Male competition over access to females is central to securing breeding opportunities in many species".

> As you suggested, we have combined the first two sentences in the Introduction as follows:

L49-50: “Male competition over access to females plays important roles to secure breeding opportunities in animals [1,2].”

3. I would replace the word "animal" with "social" at L65

> As you suggested, we have replaced “animal” with “social” at L59.

4. The paragraph from L65-L85 is great!! Really concise but clear explanation of the knowledge gap your study is filling.

> Thank you for your helpful comments and warmful words.

5. I would revise the sentence at L89-90 to read "There is evidence of paternal kin-biased behavior in primates" for clarity.

> We have revised as you suggested (L82-83).

6. I would revise the sentence starting at L102 to read something like "Japanese macaques (Macaca fuscata) provide an interesting opportunity for the investigation of paternity success of non-resident males and their contributions to the production of within-group kin-dyads"

> We have revised as you suggested (L95-96).

7. L104 Is "bisexual" really the word you're looking for? If you're referring to multiple males and females mating with each other, you might be looking for the word "polygynandrous".

> As you suggested, we have used the term “polygynandrous” at L98.

8. I would generally rephrase L104 to read "They form female-philopatric multi-male and multi-female groups characterized by strong matrilineal affiliative relationships and a polygynandrous mating system"

> We have revised as you suggested (L97-98).

9. L110 "Noteworthily" reads a bit odd to me. I would remove.

> We have deleted “noteworthily” as you suggested (L103).

10. Has a high percentage of EGPs been previously recorded for Japanese macaques specifically? Or is L110-111 referring to other species?

> Yes, a high percentage of EGPs has been previously reported (Hayakawa 2008), whereas the EGP has not been assessed at an individual level.

11. Thank you for expanding table 1 :)

> We appreciate for your suggestion.

12. Methods overall are greatly improved and offer better detail and clarity!

> We appreciate for your helpful guides.

13. I have no comment on genetic methods as this is not my area of expertise.

> Ok, but thank you for reading even genetic methods.

14. Very interesting results. The discussion appropriately contextualizes the results within the limitations of the study. I appreciate the comparisons to other specific studies and species.

> We appreciate for your helpful guides and warmful words.

Great work, authors!

> Thank you very much!

Reviewer #2: Thank you for the opportunity to review this interesting manuscript. The authors set out to examine the reproductive success of non-resident:resident male Japanese macaques in a provisioned, habituated population. There is little information on non-residents in the published literature, so this represents a unique and important contribution to the literature. Additionally, the authors calculated the number of paternal siblings in the study group, which will aid in our understanding of how kinship influences social behavior. The authors have revised the manuscript and addressed the previous two reviewers' concerns in my judgement. I recommend publication following minor revision. I used track changes in the attached document to help the authors find places where I changed sentences or had questions regarding their meaning. I wish them the best of luck with future work at this site.

> We sincerely thank your review and helpful comments on our manuscript. Based on your comments in the attached document, we have revised our manuscript. We have responded to your comments as follows:

[A1]: I suggest this phrasing, because it is the variation in RS that is being explained.

> We have revised as you suggested (L52).

[A2]: I am unsure of your meaning here--I think you mean that you did not systematically collect behavioral data from these individuals.

> Yes, as you suspected, we meant that we did not record the presence of group members systematically. To clarify the meaning here, we have changed the term “quantitatively” to “systematically” (L130).

[A3]: You may explain this later, but are the NR males the same individuals across each of your study years? Or are new males coming as NR each mating season?

> As you suggested, the information about genetic sampling should be placed later. We have thus moved the sampling information later as follows:

L196-197: “Since we collected genetic samples for 19 candidate fathers, which included three of the four identified non-resident males and accounted for 60–70% of all candidate fathers”

We considered that the information about the number of non-resident males identified should remain here. Therefore, we have revised this part as follows:

L153: “We identified four non-resident males during the study period.”

There were both types of non-resident males. Several non-resident males came across years. Several new non-resident males also came in some mating seasons.

[A4]: I am unsure of your meaning here--I think you mean reproductively active?

> Yes, we meant the estrus capable females. The estrus capable females were explained at L215-217. For clarity, we have added “estrus” before “capable” at L259.

---

## [Editor Report · Decision Letter 1]

6 Aug 2024

Paternity success for resident and non-resident males and their influences on paternal sibling cohorts in Japanese macaques (Macaca fuscata) on Shodoshima Island

PONE-D-24-26360R1

Dear Dr. ISHIZUKA,

We’re pleased to inform you that your manuscript has been judged scientifically suitable for publication and will be formally accepted for publication once it meets all outstanding technical requirements.

Kind regards,

Maria Santacà

Academic Editor

PLOS ONE
---

## [Editor Report · Acceptance letter]

11 Sep 2024

PONE-D-24-26360R1 

PLOS ONE

Dear Dr. ISHIZUKA, 

I'm pleased to inform you that your manuscript has been deemed suitable for publication in PLOS ONE. Congratulations! Your manuscript is now being handed over to our production team.

Kind regards, 

on behalf of

Dr. Maria Santacà 

Academic Editor

PLOS ONE